# Magnetic Nanoparticle-Containing Supports as Carriers of Immobilized Enzymes: Key Factors Influencing the Biocatalyst Performance

**DOI:** 10.3390/nano11092257

**Published:** 2021-08-31

**Authors:** Valentina G. Matveeva, Lyudmila M. Bronstein

**Affiliations:** 1Department of Biotechnology and Chemistry, Tver State Technical University, 22 A. Nikitina St, 170026 Tver, Russia; matveeva@science.tver.ru; 2Regional Technological Centre, Tver State University, Zhelyabova Str., 33, 170100 Tver, Russia; 3Department of Chemistry, Indiana University, 800 E. Kirkwood Av., Bloomington, IN 47405, USA; 4Department of Physics, Faculty of Science, King Abdulaziz University, P.O. Box 80303, Jeddah 21589, Saudi Arabia

**Keywords:** metal oxide, nanoparticle, magnetic, immobilized enzymes, biocatalyst

## Abstract

In this short review (Perspective), we identify key features of the performance of biocatalysts developed by the immobilization of enzymes on the supports containing magnetic nanoparticles (NPs), analyzing the scientific literature for the last five years. A clear advantage of magnetic supports is their easy separation due to the magnetic attraction between magnetic NPs and an external magnetic field, facilitating the biocatalyst reuse. This allows for savings of materials and energy in the biocatalytic process. Commonly, magnetic NPs are isolated from enzymes either by polymers, silica, or some other protective layer. However, in those cases when iron oxide NPs are in close proximity to the enzyme, the biocatalyst may display a fascinating behavior, allowing for synergy of the performance due to the enzyme-like properties shown in iron oxides. Another important parameter which is discussed in this review is the magnetic support porosity, especially in hierarchical porous supports. In the case of comparatively large pores, which can freely accommodate enzyme molecules without jeopardizing their conformation, the enzyme surface ordering may create an optimal crowding on the support, enhancing the biocatalytic performance. Other factors such as surface-modifying agents or special enzyme reactor designs can be also influential in the performance of magnetic NP based immobilized enzymes.

## 1. Introduction

Magnetic nanoparticle (NP)-containing supports for catalytic and biocatalytic systems have received considerable attention as they allow for significant improvements in process intensification, as well as savings in both energy and materials due to magnetic recovery [1,2,3,4,5,6,7,8,9,10,11,12,13]. In batch processes, magnetic separation affords easy separation of the catalyst and further reuse. In flow processes, a magnetic catalyst can be fixed in the reactor with an applied magnetic field, or can be washed off when the magnetic field is turned off [14,15]. In this review, we are focusing on biocatalysts, i.e., immobilized enzymes, supported on magnetic carriers, although the relationship between the catalyst properties and the characteristics of the magnetic support could be quite general for both conventional catalysts and biocatalysts. The first reviews on immobilized enzymes were published by several groups in the early and late seventies [16,17,18,19], with Chibata giving definition to immobilized enzymes as biocatalysts [16]. Since then, numerous studies have been reported on the applications of immobilized enzymes in the catalysis of organic reactions as biosensors, for wastewater treatment, for enzyme essays, etc. [20,21,22,23]. To the best of our knowledge, the first enzyme immobilization on magnetic NPs was reported by de Cuyper in 1992 [24]. From 2003, magnetic NP supports for the immobilization of enzymes have been widely utilized [25,26,27], with a growing sophistication of biocatalytic systems [28,29,30].

Clearly, for all these systems, magnetic recovery is a straightforward advantage which is commonly discussed in literature due to the facilitated biocatalyst reuse. Two major design features of magnetic biocatalysts are reported. In one case, magnetic NPs are isolated from the biocatalytic species. In the other case, they are in the vicinity of enzyme molecules. The latter design feature could become a crucial factor if the bioactivity of immobilized enzymes is influenced by the presence of magnetic NPs, normally consisting of metal oxide or ferrite. In this case, however, attention needs to be paid to the pH of the reaction solution because metal oxide or ferrite NPs are digested by acids, so the biocatalyst will lose its magnetic character. Finally, we will discuss other important parameters which could significantly influence the biocatalytic performance, whether magnetic NPs are isolated from enzymes (by a silica shell, polymer, etc.), or exposed to the reactants. One of the most influential parameters is hierarchical porosity, where pore sizes can vary from micropores to macropores, providing both efficient mass transfer in the biocatalytic reaction, and a proper positioning of enzymes inside the pore for maximized activity. The parameters, such as an addition of surface-modifying agents or utilizing special enzyme reactor designs, can be also crucial for the performance of magnetic NP-based immobilized enzymes.

Due to the above advantages, studies of magnetic NP-based biocatalytic systems have enjoyed an intense growth in the last decade. From 2016 to date, 523 papers have been published which show the publication profile displayed in Figure 1; although the number of publications in 2020, and even in 2021, could be misleading because of the COVID-19 pandemic, accompanied by a slowing of scientific research.

In this review, we will discuss magnetic NP-containing supports for immobilized enzymes from the viewpoint of magnetic separation, and an enhancement of the biocatalytic activity due to the presence of oxide or ferrite NPs, surface modifiers, etc., following these trends from 2016 through to July of 2021.

## 2. Magnetic Recovery as a Major Advantage

The magnetic recovery of immobilized enzymes is normally realized by tethering enzymes to magnetic NPs via some linkers such as polymers, difunctional molecules, etc. [31,32,33,34,35,36,37]. To create an appreciable magnetic moment for a successful separation, magnetic NPs are often organized into clusters [34,38], imbedded in gels [36], other polymer or inorganic matrices [13,39,40,41], or self-assembled into larger structures [42,43]. This creates interparticle interactions and increases magnetic attraction between magnetic NPs and an external magnetic field. In the case of polymers utilized in the biocatalyst development, NPs can be formed separately and then mixed with the polymers, as is most frequently done for cellulose, because of its limited solubility [33]. The other avenue in the magnetic NP-polymer composites is realized when macromolecules form a brush tethered to the NP surface [34] or a polymer shell using in situ polymerization [44]. For a thermoresponsive polymer brush [34], the terminal functionality of macromolecules has been employed to attach an enzyme, while the responsive character of the brush allowed for control of enzymatic activity (Figure 2). Clustering of the NPs surrounded by the thermoresponsive polymer allows for easy magnetic separation.

Finally, the formation of magnetic NPs in the presence of polymers is often preferred because it simplifies the biocatalyst preparation process [30,45].

## 3. Enhancement of Enzymatic Activity

The enhancement of enzymatic activity in magnetic biocatalysts can be achieved in several ways, including due to magnetic NPs or organic modifiers [15,46]. An especially fascinating enhancement of the enzyme activity was observed when both enzyme and iron oxide NPs were located in the vicinity of each other. The comparison of the activities of glucose oxidase (GOx) immobilized in porous silica (SiO_2_), alumina (Al_2_O_3_), and zirconia (ZrO_2_), or in the same supports but containing magnetite NPs in the support pores (Fe_3_O_4_-SiO_2_, Fe_3_O_4_-Al_2_O_3_, Fe_3_O_4_-ZrO_2_), revealed that biocatalysts based on magnetic supports are noticeably more active than those based on non-magnetic supports [13,41]. Figure 3 shows a trend for ZrO_2_ and Fe_3_O_4_-ZrO_2_ supports [41]. This effect was assigned to the inherent enzyme-like activity of iron oxide NPs [47,48,49], which leads to an enhancement of the activity of immobilized GOx due to a synergetic effect. In this case, Fe_3_O_4_ serves as a co-catalyst for the enzyme, increasing its activity [50].

Suo et al. reported a considerable increase of the activity of the lipase based magnetic biocatalyst due to ionic liquids, whose presence improved the microenvironment of the immobilized lipase by decreasing the support hydrophobicity [37]. Furthermore, it modified the secondary lipase structure and allowed for the exposure of the enzyme active site. Crosslinking of the chitosan-magnetic NP support with immobilized xylanase and filter paper-ase using genipin allowed for a significant enhancement of activity compared to a non-crosslinked biocatalyst [31]. This was attributed to a higher local concentration of the substrate in the confined space, resulting in a boosted number of interactions between the substrates and immobilized enzymes.

## 4. Porosity Influence on the Biocatalyst Performance

Magnetic porous supports were prepared by combining magnetic NPs and inorganic [13,41,51], polymeric [51,52] or carbon [53] porous materials. Magnetite NPs combined with reduced graphene oxide (rGO) plates were utilized as precursors for spray pyrolysis fabrication of nearly spherical porous magnetic supports for high efficiency enzyme immobilization [53]. Among porous polymers, thermoresponsive polymers received considerable attention [52,54]. Shen et al. designed a controllable thermoresponsive membrane formed by a block copolymer with imbedded magnetite NPs and immobilized enzymes, whose performance was tested at varying temperatures between 25 and 39 °C [52]. A swollen hydrophilic state at the high temperature allowed for better interactions of the substrate with the enzyme, thus allowing high biocatalytic activity of the enzyme reactor (Figure 4).

In many cases, magnetic porous supports are created by placing a porous shell on a magnetic core [55,56,57,58,59], allowing for the isolation of magnetic NPs from the part of the porous shell where enzymes are immobilized. This is also a traditional path to prevent magnetic NP aggregation, or their influence, on the enzymatic behavior. An original approach was realized in creating pod-like 1D structures by self-assembly of core-shell Fe_3_O_4_-SiO_2_ NPs under an applied magnetic field followed by the formation of mesoporous silica in the presence of a surfactant—cetyltrimethylammonium bromide (CTAB) (Figure 5) [60]. The material is characterized by a tunable hollow space and vertical pores of 8.2 nm, exceeding the size (4 nm) of the immobilized lipase. This design allowed a high loading capacity and enhanced catalytic activity.

Across numerous studies, hierarchical porosity of biocatalyst supports has been shown to be crucial in influencing the biocatalyst performance [57,58,61,62,63,64,65,66]. The advantages of hierarchical porosity are threefold: (i) small pores (micropores) provide structural integrity of the support, (ii) large pores (large meso- and macropores) allow an improved mass transfer of the substrate, (iii) while the medium-sized pores (mesopores, which are larger than the enzyme size) can result in optimal self-assembly of enzyme molecules inside the pores, resembling the degree of crowding realized in cells. This factor could profoundly increase the activity of immobilized enzymes [67]. Below we discuss a few examples of magnetic biocatalysts with hierarchical porosity.

Magnetic microspheres with hierarchical porosity (PFMMs) were fabricated using a non-conventional precursor—a novel rigid-flexible dendrimer synthesized by interfacial polymerization of trimesoyl chloride (TMC) and 1,6-hexanediamine (HDA) (Figure 6) [66].

PFMMs possess pore sizes in the range of 5–75 nm and excellent loading capacity for covalent immobilization of *Pseudomonas fluorescens* lipase (PFL). The increased rigidity of this support compared to that of the support based on fully flexible dendrimers also allowed an improved reusability of the immobilized PFL.

Magnetic metal-organic frameworks (MOFs) containing micro-, meso-, or even macropores demonstrated a promise as supports for enzyme immobilization due to a combination of porosity, magnetic separation, and pore ordering, thus allowing for more ordering in the enzyme positioning [68,69]. Usually in hierarchical MOFs, mesopores are formed in microporous frameworks and, thus, larger pores are connected by micropore channels, impeding accessibility for large substrates. Lu et al. used a different approach, i.e., forming microporous MOFs in continuous mesoporous tunnels [70]. In the case of the magnetic imidazolate framework (ZIF-8) containing a highly ordered macroporous structure, catalase molecules were immobilized both outside and inside of macropores (due to an average pore size of 69 nm), and allowed a three-fold increase of the enzyme loading capacity compared to conventional ZIF-8 (consisting of solely micropores) as well as a much higher stability [68]. Hierarchically porous Fe-MOFs were grown on the surface of Fe_3_O_4_ NPs using a solvothermal method, and employed for immobilization of chloroperoxidase or horseradish peroxidase [57]. The behavior of these biocatalysts in the degradation of organic toxins revealed that decreased diffusion resistance of substrates (due to their concentration on the hierarchically porous support near enzymes) enhances the biocatalyst efficiency.

Another novel magnetic hierarchically porous MOF was prepared via a modulator-induced defect-formation strategy (Figure 7) [63]. Polydopamine (PDA) was utilized as a source of amino groups to coordinate Zr^4+^, followed by the removal of dodecanoic acid (DA, a competitive ligand) using HCl, to form comparatively large mesopores. The resultant material exhibits a well-defined core-shell structure, hierarchical porosity (micropores and mesopores), and strong magnetic responsiveness. After immobilization of amidase with a high loading on this magnetic carrier, the biocatalyst showed higher efficiency, thermal stability, stability upon storage, reusability, etc. compared to the native enzyme or the analogous catalyst without hierarchical porosity.

## 5. Surface Modifying Agents

The surface modification depends on the method of the magnetic support preparation and on the components of the biocatalyst [30]. There are two major objectives in modifying the surface of magnetic supports. One is focused on the functionalization of the supports with such groups as aldehyde, amine, diimide, carboxyl, hydroxyl, etc., for further attachment of enzymes and other modifying molecules [71,72,73,74,75,76,77,78,79]. Functional groups can be provided via the attachment of difunctional molecules, polymers, dendrimers, or oligomers (aptamers) [80,81,82,83]. An interesting strategy has been proposed by Song et al. using toehold-mediated DNA strand displacement for the immobilization of enzymes on the magnetic support [84]. To accomplish this, the enzyme was conjugated with a target DNA to replace the captured DNA, modifying the magnetic support. This results in the protection of the enzyme from denaturation and leakage after multiple uses. A combination of the polyamidoamine dendrimer and DNA directed immobilization of the trypsin and created a biocatalyst with high reusability and stability [80].

A polyhistidine-tag (His-tag) has been utilized for modifying the surface of “sea-urchin” shaped NPs composed of the Ni silicate shell and the magnetite core [59]. Such a modification was possible due to the high affinity of His-tag to Ni ions. At the same time, His-tag easily attached tobacco etch virus (TEV) protease, enhancing compatibility between the support and the enzyme (Figure 8).

The other objective of the surface modification deals with tuning the hydrophobicity–hydrophilicity balance for better compatibility of the enzyme, support, and the substrate [73,76,77,83,85,86]. Lipase is a typical enzyme requiring surface modification due to its high affinity to hydrophobic molecules. In some cases, an enzyme (not a support) is modified, as has been reported for porcine pancreas lipase (PPL) [76]. PPL was functionalized with dodecyl aldehyde and combined with a magnetic support, resulting in the enzyme immobilization loading of about 100% and the biocatalyst efficiency of ~80%. In a reverse strategy, magnetic NPs coated with polymers containing long alkyl chains (octyl or hexadecyl) have been utilized for immobilization of *Candida rugosa* lipase (CRL) [82]. Hexadecyl tails allowed for higher loading of the enzyme and its binding in the open conformation, promoting enzyme performance. On the other hand, very hydrophobic multiwall carbon nanotubes containing magnetic Co NPs required an additional functionalization with aminated polydopamine for immobilization of CRL via a covalent attachment with glutaraldehyde [75]. This biocatalyst displayed high efficiency, stability in a wide pH and temperature range, and enhanced reusability. Here, mere hydrophobicity without functionality was not sufficient for CRL immobilization.

Functional ionic liquids have been reported as modifying agents for lipase immobilization on a magnetic support [85,86]. The authors coated magnetite NPs with chitosan (CS), attached an imidazole-containing ionic liquid (IL), and then immobilized PPL (Figure 9) [85]. The enhanced immobilization degree, biocatalyst efficiency, as well as higher tolerance to pH and temperature changes were assigned to improved biocompatibility due to CS and the effect of the IL ion, protecting the PPL native conformation.

## 6. Conclusions

In this short review, we analyzed the major advantages of magnetic supports for enzyme immobilization, and the influence of their structural features on the biocatalytic properties. The most straightforward advantage of magnetic biocatalysts is the easy separation due to magnetic forces between magnetic NPs and an external magnetic field, facilitating the biocatalyst reuse. It allows for savings of energy, materials, and time, facilitates reuse, and results in less expensive processes, paving the way for future industrial applications. Another remarkable benefit is realized when enzymes are attached in close proximity to iron oxide NPs, thus allowing for synergy between the enzyme and iron oxide NPs due to the inherent enzyme-like activity of the latter. We believe this enhancement is underutilized because, very frequently, iron oxide NPs are isolated from enzymes by silica or polymer shells with the assumption that magnetic NPs could be detrimental for biocatalyst performance. Indeed, this could be the case if the media is acidic (very rarely for biocatalytic reactions) or if iron oxide promotes a side reaction. We believe that for all other cases, co-existence of immobilized enzymes and iron oxide NPs in proximity to each other is highly beneficial.

The other major benefit of magnetic biocatalysts is the skillfully developed hierarchical porosity of the supports, allowing a controlled enzyme immobilization for the design of efficient biocatalysts. The advantages of hierarchical porosity include easy mass transport of reacting molecules in large pores, optimal self-assembly of enzyme molecules inside the medium size pores (resulting in an enhancement of the enzyme activity), and preservation of structural integrity of the biocatalyst due to small pores. Magnetic MOFs grant an additional advantage of pore ordering.

Finally, surface modifying agents play an important role in the enhancement of the biocatalytic performance, for example, in the case of hydrophobic lipases, allowing one to adjust the support or lipase hydrophobicity, and to improve compatibility between the magnetic support, enzyme, and reacting molecules. We believe a combination of the advantages of magnetic nanoparticles (magnetic separation and enzyme-like properties) with well-designed porosity and targeted surface modification is a promising avenue for the successful development of efficient magnetic biocatalysts.

## Figures and Tables

**Figure 1 nanomaterials-11-02257-f001:**
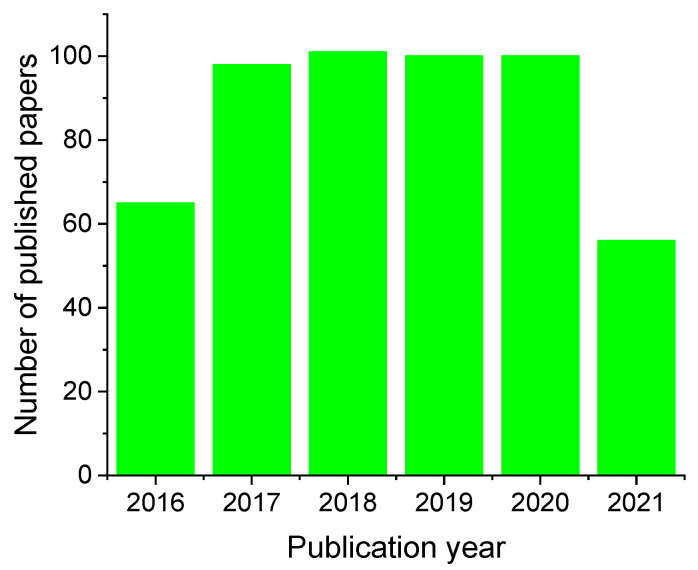
Number of publications per year in the English language. (SciFinder Search with keywords: immobilized enzymes + magnetic nanoparticles on 28 July 2021).

**Figure 2 nanomaterials-11-02257-f002:**
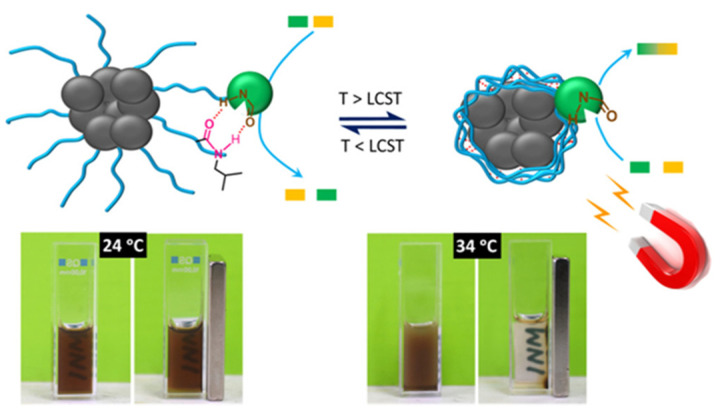
Schematic representation of the biocatalyst formed from magnetic NP clusters coated with a thermally responsive brush with attached enzyme. LCST stands for the lower critical solution temperature, while INM is the abbreviation of the Institute for New Materials (authors’ affiliation). Reproduced with permission from [34], the American Chemical Society, 2020.

**Figure 3 nanomaterials-11-02257-f003:**
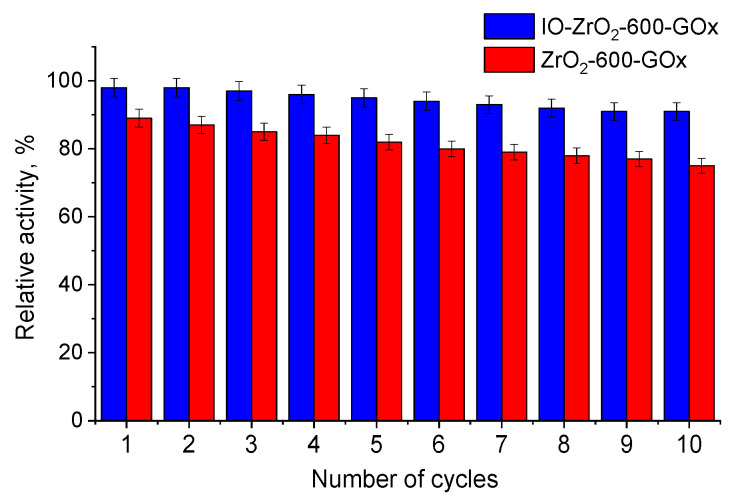
Relative activity of GOx immobilized on ZrO_2_ or Fe_3_O_4_-ZrO_2_ in reuse. IO stands for Fe_3_O_4_ NPs, while 600 indicates the temperature at which ZrO_2_ was treated. Reproduced with permission from [41], the American Chemical Society, 2020.

**Figure 4 nanomaterials-11-02257-f004:**
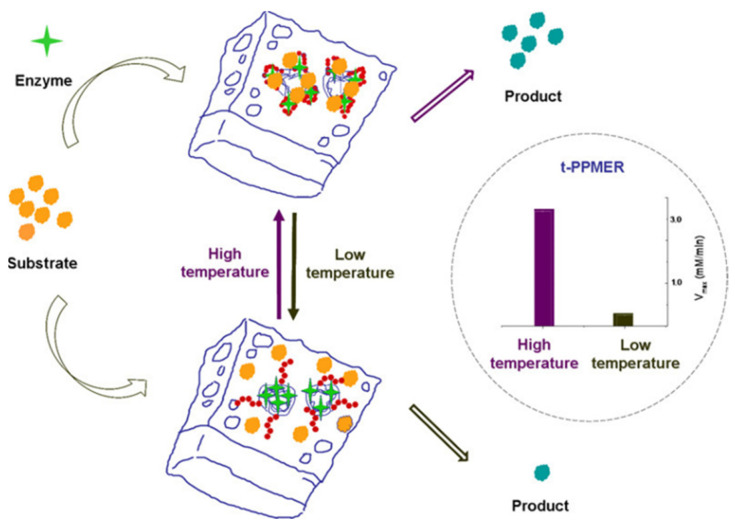
Schematic representation of thermoresponsive behavior of the biocatalyst. T-PPMER stands for thermoresponsive porous polymer membrane enzyme reactor. Reproduced with permission from [52], the American Chemical Society, 2021.

**Figure 5 nanomaterials-11-02257-f005:**
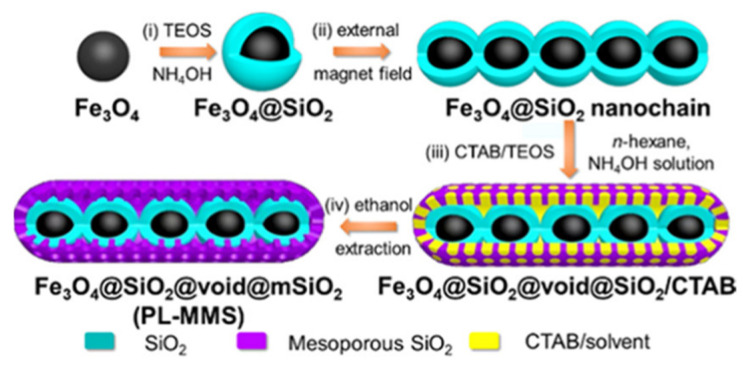
Synthesis Procedures of PL-MMS Nanochain: (i) Sol–Gel Coating of Silica; (ii) Magnetic-Field-Guided Assembly; (iii) Mesostructured CTAB/Silica Interface Assembly onto the Nanochain; (iv) Surfactant Removal. Reproduced with permission from [60], the American Chemical Society, 2020.

**Figure 6 nanomaterials-11-02257-f006:**
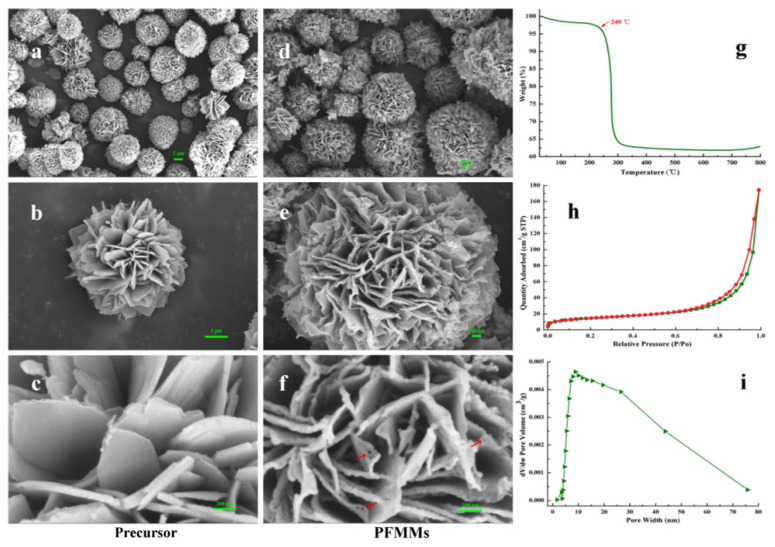
SEM micrographs and properties of PFMMs and its precursor. (**a**–**c**) PFMMs precursor. Their scale bars are 2 μm, 1 μm, and 200 nm, respectively. (**d**–**f**) PFMMs. Their scale bars are 1 μm, 1 μm, and 200 nm, respectively. (**g**) TGA curve of PFMMs precursor. (**h**) N_2_ adsorption–desorption isotherm. (**i**) Pore size distribution of PFMMs. Reproduced with permission from [66], the American Chemical Society, 2020.

**Figure 7 nanomaterials-11-02257-f007:**
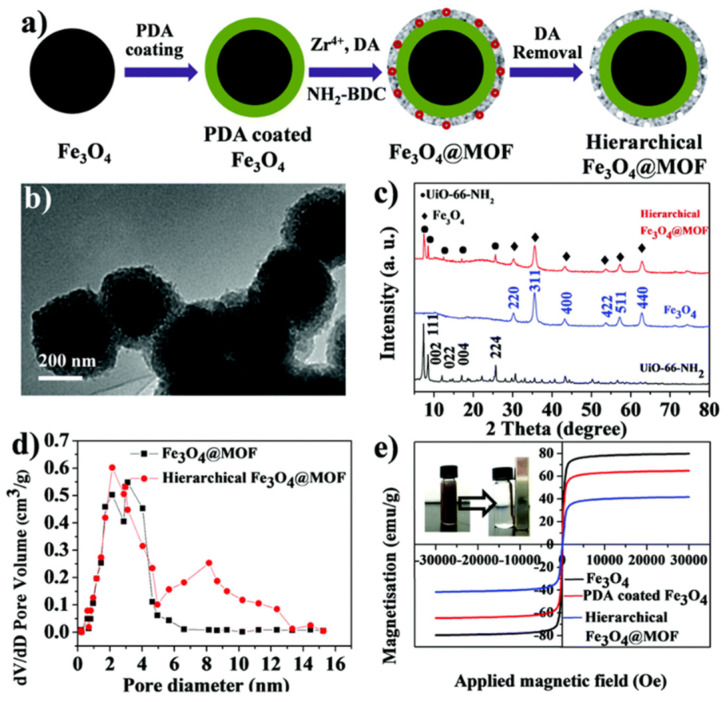
Synthesis and characterization of the magnetic hierarchically porous MOF (hierarchical F_3_O_4_@MOF). (**a**) Schematic illustration for the three-step preparation; (**b**) TEM image of the hierarchical F_3_O_4_@MOF; (**c**) XRD profiles of UiO-66-NH_2_ (MOF, bottom), F_3_O_4_ (middle), and hierarchical F_3_O_4_@MOF (top); (**d**) pore size distribution curves of F_3_O_4_@MOF (black) and hierarchical F_3_O_4_@MOF (red); and (**e**) magnetic hysteresis loops of F_3_O_4_, PDA coated F_3_O_4_ and hierarchical F_3_O_4_@MOF, inset is the photographs of the magnetic responsiveness. Reproduced with permission from [63], the Royal Society of Chemistry, 2019.

**Figure 8 nanomaterials-11-02257-f008:**
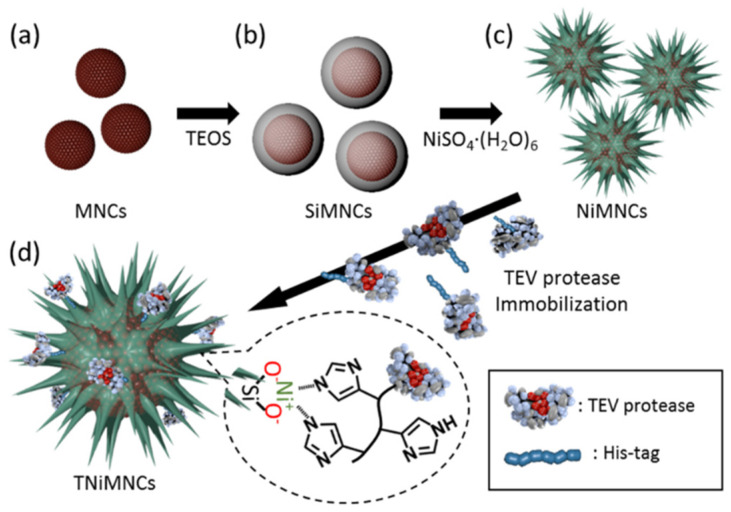
Synthesis of the NiMNCs and TEV protease immobilization. (**a**) Superparamagnetic iron oxide magnetic nanocluster (MNCs) was prepared by thermal decomposition method; Formation of (**b**) silica shell on the surface of MNCs (SiMNCs) (TEOS stands for tetraethoxysilane); (**c**) Silica shell transferred to nickel silicate (NiMNCs); and (**d**) His-tagged TEV protease immobilized on the surface of NiMNCs (TNiMNCs). Reproduced with permission from [59], IOP Publishing, 2016.

**Figure 9 nanomaterials-11-02257-f009:**
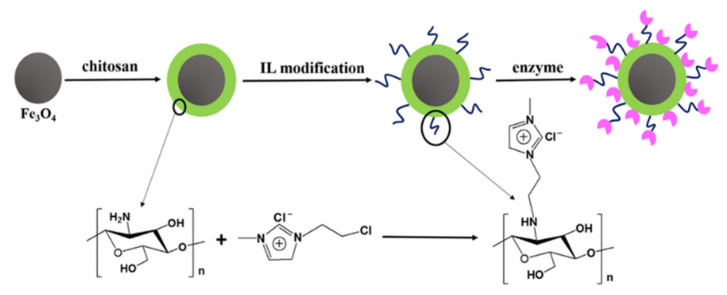
Synthetic scheme of IL-CS-Fe_3_O_4_ and its application in PPL immobilization. Reproduced with permission from [85], Elsevier, 2019.

## Data Availability

Not applicable.

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
