# Peer review of "Magnetic Nanoparticle-Containing Supports as Carriers of Immobilized Enzymes: Key Factors Influencing the Biocatalyst Performance"

_nanomaterials, 2021, doi:10.3390/nano11092257_

Round 1

Reviewer 1 Report

This review article by Matveeva and Bronstein summarizes the progress on the immobilized enzymes on the magnetic nanoparticles. 

It is a rather short review and this reviewer has the doubt whether the authors have indeed included sufficient references. This article falls short on giving the readers the historic context of the topics discussed in this article. 

In addition, the authors need to take care of the following specific issues:

1) Figures 2, 4, 5, 6, 7, 8 and 9 were adapted from other publications but they have really low resolutions in this manuscript. Please use high-resolution images. 

2) Figure 1 would need to give the exact date of collecting these numbers in the caption. For example July 15, 2021.

Author Response

Comment 1: It is a rather short review and this reviewer has the doubt whether the authors have indeed included sufficient references. This article falls short on giving the readers the historic context of the topics discussed in this article.

Response 1: Indeed, this is a short review, we call it Perspective, as was indicated in the Abstract. Nevertheless, it cited 73 references and we believe it is sufficient for this type of manuscript. Perspective is not supposed to give a historic context, but rather a modern state of the field and a its possible further development. The Abstract and the end of the Introduction clearly outline the goals of this work.

Comment 2: Figures 2, 4, 5, 6, 7, 8 and 9 were adapted from other publications but they have really low resolutions in this manuscript. Please use high-resolution images. 

Response 2: Please note that high resolution images were uploaded from the corresponding journal websites. The images look sharp in the Word document so probably reviewer’s impression is due to loss of resolution in the PDF file.

Comment 3: Figure 1 would need to give the exact date of collecting these numbers in the caption. For example July 15, 2021.

Response 3: The date of the data collection has been added in the Figure 1 caption.

Reviewer 2 Report

This paper tries to be a very short review of the last five years research of biocatalysis activity in presence of magnetic nanoparticles. This is one interesting issue but the form to study here seems to be a little poor. For instance, they refer to Suo et al. with references [15,33] for saying that the biocatalyst enhancement using the magnetic nanoparticles is due to confined ionic liquids, but this is very difficult to understand because the electric interaction is much higher than the magnetic one and in fact they only change a little bit their trayectories without any important interaction. On the other hand we need to know more details as the distribution of the ions or the size and composition of such nanoparticles, etc...Finally, they need to clearly present what are the important results nowadays and also the future results expected for taking the interest of the reader.

Author Response

Comment 1. This paper tries to be a very short review of the last five years research of biocatalysis activity in presence of magnetic nanoparticles. This is one interesting issue but the form to study here seems to be a little poor.

Response 1. We disagree with the reviewer that the form of reviewing of the literature here is poor. In a short review we are unable to give detailed explanations of cited examples, but this is not our goal. Our objective is to assess most important relationships between structure of magnetic biocatalysts and their properties in a brief form.

Comment 2. For instance, they refer to Suo et al. with references [15,33] for saying that the biocatalyst enhancement using the magnetic nanoparticles is due to confined ionic liquids, but this is very difficult to understand because the electric interaction is much higher than the magnetic one and in fact they only change a little bit their trayectories without any important interaction. On the other hand we need to know more details as the distribution of the ions or the size and composition of such nanoparticles, etc.

Response 2. Actually, in the Suo et al. reference [22] we do not claim that the “enhancement using the magnetic nanoparticles is due to confined ionic liquids”. We stated that the enhancement of activity of the magnetic biocatalyst (not magnetic nanoparticles!) is due to ”the improved microenvironment by modifying the secondary lipase structure and allowing exposure of the enzyme active site”. It has nothing to do with magnetic vs electrostatic interactions. In the revised version we modified this statement to clarify where the enhancement is coming from (page 3, lines 92-93). Also, please note that in this section we only indicate the fact of the catalyst activity enhancement in the presence of ionic liquids. On page 8 we devoted a whole paragraph to modification of immobilized lipases with ionic liquids, describing two more papers from the same group (refs. [72, 73]) and presenting Figure 9.

Comment 3. Finally, they need to clearly present what are the important results nowadays and also the future results expected for taking the interest of the reader.

Response 3. We believe Conclusions clearly identify the important recent results in the field of immobilized enzymes on magnetic supports and indicate where the field should be going, according to our opinion. This is the mission of Perspective. However, considering that the reviewer find it unclear, the text of Conclusions has been revised.

Round 2

Reviewer 1 Report

The authors made arguments in order not to do improvement on this manuscript. 

One thing which I'm still concerned about is the authors' view about "perspective" article. No matter what type of article it is, the readers should be informed about the background and historic context. It's up to the authors whether or not to do so to serve the readers in a better way.  

Author Response

Reviewer 1

Comment 1: One thing which I'm still concerned about is the authors' view about "perspective" article. No matter what type of article it is, the readers should be informed about the background and historic context. It's up to the authors whether or not to do so to serve the readers in a better way.  

Response 1: We added historical background with references in the Introduction (lines 41-53).

Reviewer 2 Report

Obviously a review is a very difficult task wheather it is long or short, because it needs to explain the progress of an issue in a very comprenhensive and attractive form. Thus one needs to see clearly what is the phenomena and that in this case the magnetic nanoparticles must do it interacting with other materials and what is the progress made so far. In my case it seems that I have failed to get it and what is worse, I follow in the same state even after this first reviewing.
Unfortunately the answers to my questions are not convincing. For instance, they claim in the abstract that:

A clear advantage of magnetic supports is due to facile magnetic separation, allowing savings of materials and energy in the biocatalytic process. 

that I cannot understand the facile magnetic separation without a magnetic interaction wherever it is acting. In Fig.2 they work with magnetic nanoparticles and they explain it at the begining of section 2, but in a meaningless form, at least for me. They refer to the page 8 for explaining the magnetic "separation" but this is practically only a sentence and a mention to references ( 72-73). This doesen't seems to be enough for a serious work that deserve the interest of the readers. 

Let me repeat my fundamental question. They write in page 3 something which needs at least, from line 101-103 where they comment about the enzime activity enhancement in presence of NP. Why? What is the reason of such behaviour?

Author Response

Reviewer 2

Comment 1: Obviously a review is a very difficult task wheather it is long or short, because it needs to explain the progress of an issue in a very comprenhensive and attractive form. Thus one needs to see clearly what is the phenomena and that in this case the magnetic nanoparticles must do it interacting with other materials and what is the progress made so far. In my case it seems that I have failed to get it and what is worse, I follow in the same state even after this first reviewing.
Unfortunately the answers to my questions are not convincing. For instance, they claim in the abstract that:

A clear advantage of magnetic supports is due to facile magnetic separation, allowing savings of materials and energy in the biocatalytic process. 

that I cannot understand the facile magnetic separation without a magnetic interaction wherever it is acting.

Response 1: We are sorry that we failed to convince this reviewer, but it is probably merely misunderstanding. We agree with the reviewer that magnetic separation is DUE TO magnetic forces between magnetic nanoparticles and an external magnetic field. In the revised manuscript we added this information in the Abstract (lines 17-18), Introduction (line 55), Section 2 (lines 91-93 and 100-101) and in Conclusion (lines 284-285).

Comment 2: In Fig.2 they work with magnetic nanoparticles and they explain it at the begining of section 2, but in a meaningless form, at least for me.

Response 2: At the beginning of section 2 we list strategies which allow magnetic separation, namely, clustering of nanoparticles, self-assembly, etc. for an increase of interparticle interactions leading to an enhanced magnetic response. We added a sentence (lines 91-93) to clarify this phenomenon. Apparently, we also failed to emphasize how magnetic separation is achieved in the case, illustrated in Fig. 2. To accomplish that, we added a sentence preceding Fig. 2 (lines 100-101).

Comment 3: They refer to the page 8 for explaining the magnetic "separation" but this is practically only a sentence and a mention to references ( 72-73). This doesen't seems to be enough for a serious work that deserve the interest of the readers. 

Response 3: Sorry but it is a misunderstanding. In this section we discuss surface modifying agents, including ionic liquids. Magnetic separation is due to magnetic nanoparticles and applicable for all the cases discussed in this review. In addition, we discuss other aspects of magnetic biocatalysts and the factors influencing their performance, which could be irrelevant to magnetic nanoparticles if they are coated with silica or polymers.  In the revised manuscript we restructured section 3 (swapped paragraphs) to emphasize that an enhancement of the biocatalyst performance can be due to magnetic nanoparticles and/or organic molecules.

Comment 4. Let me repeat my fundamental question. They write in page 3 something which needs at least, from line 101-103 where they comment about the enzime activity enhancement in presence of NP. Why? What is the reason of such behaviour?

Response 3: It is known that iron oxide nanoparticles display enzyme-like activity in a number of enzymatic reactions (see refs. 47-50 in the revised manuscript). When enzyme is attached in close vicinity to an iron oxide nanoparticle, the latter serves as a co-catalyst for the enzyme, leading to SYNERGY of their action. This is explained in the paragraph before Fig. 3, which was revised to clarify the matter (see lines 111 and 118-121).